# Exchange interaction in short lived flavine adenine dinucleotide biradical in aqueous solution revisited by CIDNP and nuclear magnetic relaxation dispersion

Ivan V. Zhukov[1,2], Alexey S. Kiryutin[1,2], Mikhail S. Panov[1,2], Natalya N. Fishman[1,2], Olga B. Morozova[1,2], Nikita N. Lukzen[1,2], Konstantin L. Ivanov[1,2], Hans-Martin Vieth[1,3], Renad Z. Sagdeev[1,2], and Alexandra V. Yurkovskaya[1,2]

[1]International Tomography Center, Siberian Branch of the Russian Academy of Sciences, Novosibirsk, 630090, Russia
[2]Novosibirsk State University, Novosibirsk, 630090, Russia
[3]Freie Universität Berlin, 14195 Berlin, Germany

*Correspondence to*: Alexandra V. Yurkovskaya (yurk@tomo.nsc.ru)

Prof. Konstantin L. Ivanov passed away on the 05.03.2021 at the age of 44.

**Abstract.** Flavin adenine dinucleotide (FAD) is an important cofactor in many light-sensitive enzymes. The role of the adenine moiety of FAD in light induced electron transfer was obscured because it involves an adenine radical, short-lived with a weak chromophore. However, an intramolecular electron transfer from adenine to flavin was revealed several years ago by R. Kaptein by using chemically induced dynamic nuclear polarization (CIDNP). The question whether one or two types of biradicals of FAD in aqueous solution are formed stays unresolved so far. In the present work, we revisited the CIDNP study of FAD using a robust mechanical sample shuttling setup covering a wide magnetic field range with sample illumination by a light emitting diode. Also, a cost efficient fast field cycling apparatus with high spectral resolution detection up to 16.4 T for nuclear magnetic relaxation dispersion studies was built based on a 700 MHz NMR spectrometer. Site-specific proton relaxation dispersion data for FAD show a strong restriction of the relative motion of its isoalloxazine and adenine rings with coincident correlation times for adenine, flavin and their ribityl-phosphate linker. This finding is consistent with the assumption that the molecular structure of FAD is rigid and compact. The structure with close proximity of the isoalloxazine and purine moieties is favorable for reversible light induced intramolecular electron transfer from adenine to triplet excited flavin with formation of a transient spin-correlated triplet biradical F$^{\bullet-}$-A$^{\bullet+}$. Spin selective recombination of the biradical leads to the formation of CIDNP with a common emissive maximum at 4.0 mT detected for adenine and flavin protons. Careful correction of the CIDNP data for relaxation losses during sample shuttling shows that only a single maximum of CIDNP is formed in the magnetic field range from 0.1 mT to 9 T; thus, only one type of FAD biradical is detectable. Modeling of the CIDNP field dependence provides good agreement with the experimental data for a normal distance distribution between the two radical centers around 0.89 nm and an effective electron exchange interaction of –2.0 mT.

# 1 Introduction

Flavins play an important role as coenzymes in various biological systems and therefore have been studied extensively. Thus, the optical absorption and fluorescence properties of ground state, excited states, and various free radical forms have been well characterized. Flavin adenine dinucleotide (FAD) attracted much attention in the last decades as a cofactor of the cryptochrome photoreceptor suggested to be responsible for sensitivity to the Earth's magnetic field in animal and avian navigation (Wiltschko and Wiltschko, 2019). A review on the radical-pair mechanism (RPM) of magnetoreception as a leading hypothesis to explain bird navigation can be found in the literature (Hore and Mouritsen, 2016). The same mechanism of photocycle and signaling action of plant cryptochrome in arabidopsis was reviewed recently (Ahmad, 2016). The keystone of the proposed explanation is as follows: the blue-light activated flavin moiety of FAD oxidizes a chain of three tryptophans resulting in a radical pair composed of a singly reduced semiquinone flavin and an oxidized tryptophan. Accordingly, the singlet/triplet spin dynamics of the FAD⁻/Trp+ radical pair has been intensively studied as the source of cryptochrome sensitivity to the Earth's magnetic field.

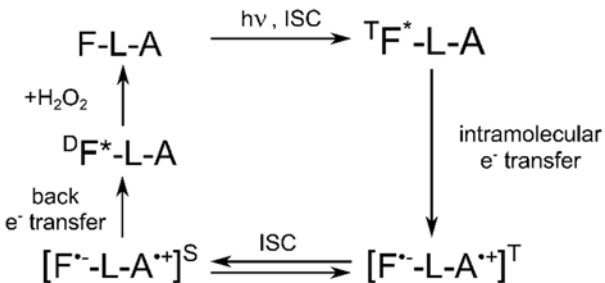

**Scheme 1 Cyclic photochemical reaction of reversible intramolecular electron transfer from adenine to flavin in FAD molecules in aqueous solution. The triplet-singlet transition in the short lived flavin-adenine biradical leads to formation of CIDNP. The FAD molecule is shown as F-L-A, F - denotes flavin, L – the ribityl-phosphate linker, and A - adenine.**

In all these studies, the role of adenine was obscure or limited to the role of a binding site to the protein, but interaction between the photo-excited flavin and adenine was neglected. As a result, the light induced reaction of intramolecular electron transfer between the two FAD moieties was not considered, presumably because the adenine radical is a weak chromophore, being hardly detectable by optical methods (Murakami et al., 2005; Antill and Woodward, 2018). To the best of our knowledge, the influence of the flavin–adenine biradical on the magnetic field dependence of transient absorption was not taken into consideration by a large scientific community, probably because most of their studies were based on various optical methods. However, as it was shown by Robert Kaptein and co-workers (Stob et al., 1989), upon light excitation of the flavin moiety of FAD a short–lived triplet biradical is formed by intramolecular electron transfer from the adenine (see Scheme 1).

In that breakthrough study, Kaptein applied chemically induced dynamic nuclear polarization (CIDNP) being a sensitive tool for magnetic resonance characterization of short lived radicals that are too elusive for EPR detection or do not have a suitable optical band. In Kaptein's work (Stob et al., 1989), CIDNP effects arising from the FAD biradical were reported at high and low magnetic field under continuous light illumination. In the field dependence of emissive nuclear polarization, two contributions were discerned with the maxima at 3.0 and 10.0 mT, respectively. Strong electronic exchange interaction was revealed, much higher than the Earth's field. This type of interaction splits the singlet and triplet states of the biradical and leads to a CIDNP maximum at the level anti-crossing of one of the triplet electronic states $T_{\pm}$ with S. The presence of avoided level crossings in the primary biradical is encoded in the magnetic field dependence of the reaction yield and, in general, in the lifetime of the flavin radical. Often FAD is discussed as a candidate molecule responsible for the formation of such spin-correlated radical pairs in living organisms that contain particular proteins - blue light photoreceptors, cryptochromes, which contain a non-covalently bound FAD photoreceptor molecule. The radical pair usually considered is a pair of radicals [FAD˙⁻ Trp˙⁺], which is formed by sequential electron transfer along the chain of tryptophan residues to the cofactor FAD in

cryptochrome (Dodson et al., 2015). However, the appearance of the magnetic field effect in this secondary radical pair, [FAD$^{\bullet-}$ Trp$^{\bullet+}$], formed in parallel or subsequently from the FAD biradical might be significantly affected by the spin dynamics in the primary FAD biradical.

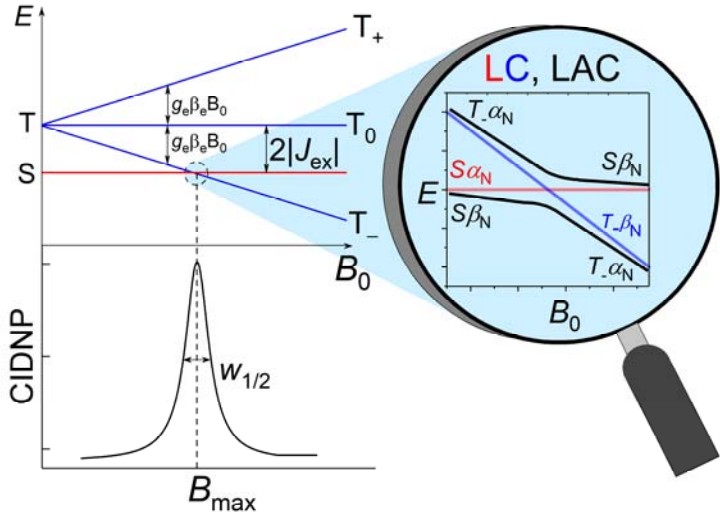

**Fig.1. Energy levels diagram of a biradical or radical pair with negative exchange interaction and the level anti-crossing (LAC) mechanism explaining the resulting dependence of CIDNP on the magnetic field.**

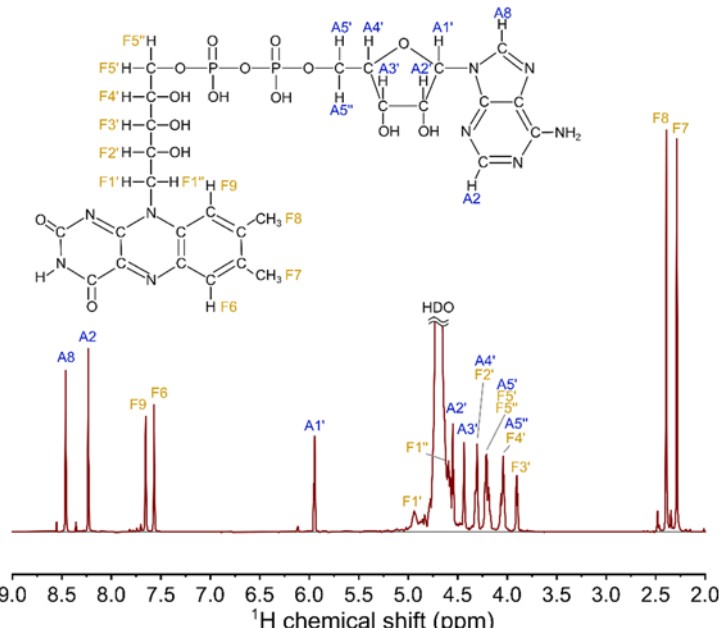

**Fig.2 700 MHz $^{1}$H NMR spectrum of 4.2 mM FAD solution in D$_2$O, pH 2.7, 25 °C.**

75 Let us briefly explain the mechanisms of such nuclear spin polarization formation in transient radical pairs or biradicals with non-zero exchange interaction. For simplicity, we consider the case of a biradical with only one spin-1/2 nucleus and an exchange interaction much larger than the hyperfine coupling (HFC) with that single nucleus. A distinct maximum ("J-resonance") of nuclear polarization in the vicinity of the level crossing (LC) between the electronic singlet and one of the triplet states ($T_+$ or $T_-$ for positive or negative J, respectively) of the biradical is detected by NMR in the diamagnetic products.

The reason is that nonsecular terms of HFC induce transitions conserving the z-projection of the total spin of the two electrons and the nucleus on the direction of the external magnetic field $B_0$. These transitions convert the LC into a level anti-crossing (LAC) (see Fig.1). For instance, when we have an $S \leftrightarrow T_-$ LC, the HFC as perturbation makes an LAC from the $S\beta_N \leftrightarrow T_-\alpha_N$ crossing but does not affect the $S\alpha_N$ and $T_-\beta_N$ levels, which thus stay uncoupled to any other states. Hereafter, the subscript '$_N$' denotes the nuclear spin state.

It was a challenging task in those days to analyze the involvement of the transient FAD biradical by means of magnetic resonance, because of short relaxation times $T_1$ of the FAD protons. For studying CIDNP at variable magnetic field a home-built falling-tube system was utilized, which allowed for sample transfer between two magnets, one used for generation of the CIDNP effect and another one for its detection, but the transfer times were in conflict with $T_1$. The open question that stays even after more than 30 years, is whether one or two biradicals with different inter-radical distance are actively contributing
to CIDNP formation. In the meantime, the frontiers of nuclear hyperpolarization methods in general, and particularly the experimental tools for CIDNP detection were considerably improved allowing to get the answer to the question. Several milestones on that way date back to Robert Kaptein. The first milestone can be attributed to the introduction of time resolved CIDNP with microsecond resolution detection for which he was among the pioneers (Hore and Kaptein, 1982). A second milestone was significant improvement of the fast field cycling technique, FFC, by introducing digitally controlled rapid
mechanical shuttling of the NMR sample over an ultra-wide magnetic field range with high resolution NMR detection (Zhukov et al., 2018). In our laboratory at the ITC in Novosibirsk, we built up such a state-of-the-art mechanical shuttle device for available 400 and 700 NMR spectrometers. Such set-ups allow one to get site-specific information about molecular mobility with atomic resolution and to run CIDNP at variable magnetic field in a fully automatic way (Zhukov et al., 2020a; Zhukov et al., 2020b). Last but not least, coherent transfer of hyperpolarization among scalar coupled spin as predicted by Kaptein (De
Kanter and Kaptein, 1979) was firmly implemented into interpretation of CIDNP formed at low magnetic fields (Ivanov et al., 2008). Armed with these important improvements, we re-examine in the present paper the former CIDNP study of FAD with the aim to refine information on involvement of the adenine radical in the short-lived primary biradical of FAD and transformation of the biradical into the secondary $FAD^-/Trp^+$ radical pair by reductive electron transfer from tryptophan to the adenine radical moiety.

**2 Materials and Methods**

Flavin adenine dinucleotide was kindly provided by Professor Kiminori Maeda from Saitama University (Japan). 99.9% $D_2O$ was purchased from Astrachem (Russia). The chemicals were used as received. The 700 MHz $^1H$ NMR spectrum of 4.2 mM FAD solution in $D_2O$, pH 2.7, 25 °C, is shown in Fig.2.

Nuclear magnetic relaxation dispersion (NMRD) experiments were run with 4.4 mM FAD solution in $D_2O$, pH 3.9, using a
110 700 MHz Bruker Avance III HD NMR spectrometer equipped with a TXI probe and a home-made fast field cycling add-on, similar to the one which has been built earlier (Zhukov et al., 2018). The 700 MHz add-on apparatus for mechanical shuttling and precise positioning NMR samples inside the warm bore of the superconducting magnet closely resembles our other set-up for the 400 MHz NMR spectrometer that was described previously (Zhukov et al., 2018). Details of the shuttling device for the 700 MHz NMR spectrometer will be published elsewhere.

The experimental protocol of the relaxation dispersion experiment is shown in Fig.3A; it is similar to the protocol used to measure nuclear magnetic relaxation dispersion (NMRD) curves of $^1H$ and $^{13}C$ nuclei of methyl propiolate (Zhukov et al., 2018). The protocol consists of 5 stages: At the first stage nuclear spins relax to equilibrium in high field, $B_0$, then a 180° pulse is applied to invert spin magnetization. Next, the sample is transferred to a position along the magnet bore where the low field

strength is equal to $B_L$. During the third stage the sample is kept in this field for a delay $\tau_{vd}$. Then in the fourth stage the sample is shuttled back to the high field $B_0$, and after application of a hard 90° RF pulse the FID is acquired. By repeating the cycle with systematically incrementing $\tau_{vd}$ we obtain a series of 1D spectra with the corresponding delays $\tau_{vd}$ in field $B_L$. Although longitudinal relaxation of nuclear spins proceeds during the whole experimental cycle, the decay of signal intensity in the NMR spectra will depend merely on the duration of relaxation delay $\tau_{vd}$, but only if the field cycling is done with sufficient reproducibility. By analyzing the decay signal intensity with $\tau_{vd}$ for various low field values $B_L$ one gets the NMRD curve - the magnetic field dependence of the longitudinal relaxation time $T_1$.

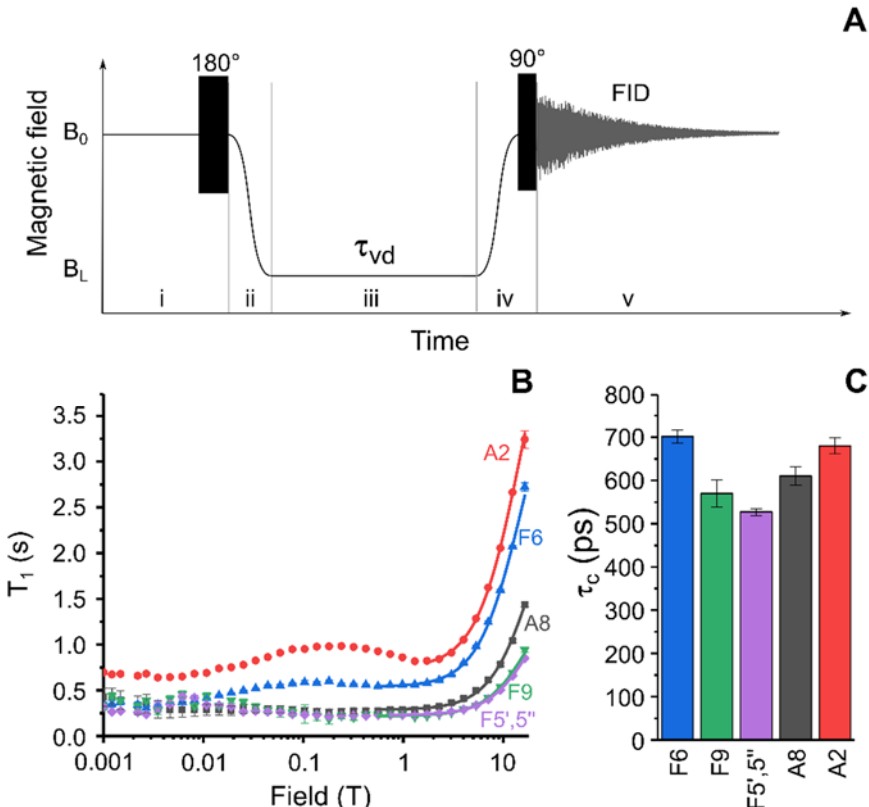

**Fig.3 A Experimental protocol for measuring nuclear spin relaxation dispersion. In step i, spins relax to thermal equilibrium at high field $B_0$, then a 180° RF pulse is applied for inversion of magnetization. In step ii, the sample is transferred to the field $B_L$, where it is kept for an incremented delay $\tau_{vd}$ during step iii. In step iv, the sample is transferred back to high field $B_0$ and a 90° RF pulse is applied. In step v, the nuclear spin free induction decay is acquired and the NMR spectrum is obtained after FT. B Relaxation dispersion data for six selected protons of FAD: A8 – black squares, A2 – red circles, F6 – blue up triangles, F5',5''– magenta diamonds, F9 – green down triangles. Lines show best fit of high-field part of relaxation dispersion curves by expression 1. C Correlation times obtained from fitting the high field part of relaxation dispersion data for protons A8, A2, F5',5'', F6 and F9.**

## 2.1 Chemically induced dynamic nuclear polarization in its dependence on the external magnetic field

CIDNP in its dependence on the external magnetic field was studied on a 400 MHz Bruker Avance III HD NMR spectrometer equipped with a fast field cycling unit and an add-on allowing for sample irradiation by compact LEDs (Zhukov et al., 2020a), (Zhukov et al., 2018). A 4 mm diameter quartz rod is used as a light guide. It is inserted into the NMR sample tube so that its polished end is positioned just above the RF coil, when the sample is placed inside the NMR probe. A 520 nm 3W LED with cooling radiator is attached to the other end of the light guide. The LED is turned on and off by an electro-mechanical relay

controlled by TTL pulses from the NMR spectrometer console in synchronization with the RF pulse sequence and the mechanical motion of the sample. To obtain CIDNP spectra, the experimental protocol depicted in Fig.6A was used: At the first stage, sample magnetization relaxes at high field $B_0$ to its equilibrium value. Then, the sample is transferred to a position with the desired magnetic field strength $B_L$, where the LED is switched on for a fixed time interval $\tau_{irr}$ = 0.5 seconds. Next, the sample is transferred back to the NMR probe at high field $B_0$ where the FID is acquired after application of a hard 90° RF pulse. For removal of thermal background polarization two spectra are recorded, one with the LED being switched on at low field, and the second one with the LED switched off. The difference between the two spectra gives the CIDNP spectrum. A typical CIDNP spectrum of the FAD sample at pH 2.7 detected at $B_L$ = 4 mT is shown in Fig.6B.

Since the sample transfer time in our 400 MHz setup is comparable with $T_1$ of FAD protons it was necessary to reconstruct the real CIDNP field dependence by deconvolution of the observed CIDNP field dependence and the proton relaxation dispersion data. For this reconstruction, the computed dependence of the external magnetic field on time passed since transfer started (sample transfer time-field profile) $B_i \rightarrow B_0$ is divided into 500 intervals $\Delta t_n > 0$, for each field value $B_i$ in the CIDNP field dependence. These intervals are counted decreasingly, so the first interval has number $n = 500$ and the last interval (just prior to FID acquisition) has number $n = 1$. For relaxation deconvolution purposes the magnetic field $B_n$ within each time interval was supposed to stay constant. Next, the measured nuclear spin relaxation dispersion curve is interpolated by cubic splines for all magnetic field values $B_n$, giving relaxation rates $R_n$ for each interval. Finally, the true CIDNP intensity $P_{true}(i)$ generated in field $B_i$ is reconstructed from the observed CIDNP intensity $P_{observed}(i)$ by the formula: $P_{true}(i) = P_{observed}(i) \cdot \prod_{n=1}^{500} \exp(R_n \Delta t_n)$. The numerical simulation shows that approximately one half of the A8 CIDNP signal is lost during sample transfer to high field.

## 2.2 Time resolved CIDNP at high magnetic field

Our setup for TR-CIDNP measurements has already been described in detail (Morozova et al., 2007). The samples purged with pure nitrogen gas and sealed in a standard NMR Pyrex ampule were irradiated in the probe of a 200 MHz Bruker DPX-200 NMR spectrometer (magnetic field 4.7 T, resonance frequency of protons 200 MHz) by laser pulses from a Brilliant B, Quantel Nd:YAG laser using its third harmonic (wavelength 355 nm, pulse length about 5 ns, output pulse energy 70-80 mJ). Light to the sample was guided using an optical system with a prism, and a light-guide quartz rod (diameter 5 mm). The TR-CIDNP spectra were obtained in the following way: (1) saturation with broadband radio frequency (RF) pulses, (2) a 10 ns laser pulse triggered by the spectrometer and (3) a detecting RF pulse of 1 µs duration followed by FID acquisition. The laser pulse was synchronized with the front edge of the RF pulse. As the background signals from Boltzmann polarization were suppressed by saturation pulses, in the CIDNP spectra only the NMR signals from the polarized products of the cyclic photochemical reaction appear.

## 3 Results and Discussion

In a FAD molecule, the adenine and isoalloxazine rings are connected with each other by a long flexible ribityl-phosphate linker, therefore it is anticipated *a priori* that the FAD molecular structure in solution is likely to be represented by a number of conformations. As an example, two extreme cases of FAD conformations are shown in Fig.4, one of them being "closed" and the other one "open". The "closed" conformation was obtained in our DFT calculations of the 3D-structure of a triplet excited FAD molecule in aqueous solution using the Gaussian program package (Frisch et al., 2009). Also, DFT calculations of IR spectra and FAD-water complex structure (Kieninger et al., 2020) have shown that the closed stacked conformation of FAD is stabilized by water molecules forming hydrogen bonds between adenine N7 in the purine ring and the ribityl chain. Another example of the extended "open" conformation was deduced from the analysis of X-ray diffraction data of the FAD-protein complex (PDB ID fda (Bruns, 1995), (Berman et al., 2000)). Moreover, in the "open" conformation a hindered rotation might happen around the single bonds connecting the adenine ring to the ribose cycle and the isoalloxazine ring to the ribityl linker.

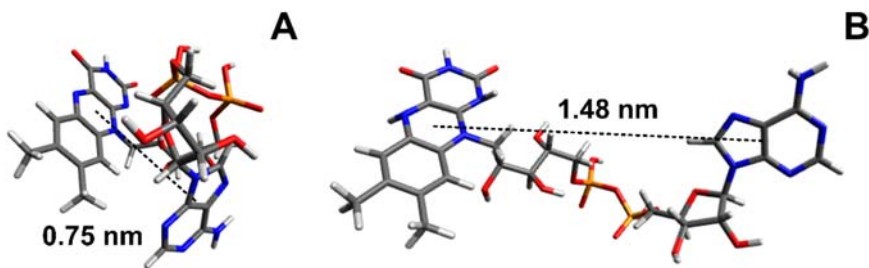

**Fig.4 A Ground-state conformation of protonated FAD calculated in the Gaussian program package (Frisch et al., 2009). B Example of stretched FAD conformation (PDB ID fda (Bruns, 1995), (Berman et al., 2000)). Pictures were made using Avogadro molecule visualizer (Hanwell et al., 2012).**

The obscured information about the preferable structure of FAD in aqueous solution is encoded by NMR in the correlation times of intramolecular mobility of individual protons and can be obtained from nuclear magnetic relaxation dispersion (NMRD), i.e. the dependence of the relaxation times on the magnetic field. With the aim to gain site specific information about correlation times, we studied NMRD of protons with high spectral resolution over a wide range of magnetic fields, from 0.1 mT to 16.4 T.

For medium-sized molecules like FAD in water at room temperature the transition between the fast and slow motional regimes, i.e. $\gamma_H B \tau_c \sim 1$, occurs in a field on the order of several Tesla, which is manifested in a characteristic increase of the longitudinal relaxation time $T_1$. To get insight into the relative mobility of the adenine and isoalloxazine rings in solution, we analyzed the NMRD curves of all protons of FAD assuming a simple empirical model with two contributions to relaxation, one involving a site-specific local field correlation time $\tau_c^i$ and another one being a field-independent constant. Accordingly, the total relaxation rate is given by the sum $R_1^{tot}(B) = \frac{R_1}{1+(\gamma_H B \tau_c)^2} + R_1^{inf}$ and

$$T_1^i(B) = \frac{1+\left(\gamma_H B \tau_c^i\right)^2}{R_1 + R_1^{inf}\left(1+\left(\gamma_H B \tau_c^i\right)^2\right)} \tag{1}$$

where $\gamma_H$ is the proton gyromagnetic ratio, $\tau_c^i$ the site-specific local field correlation time for $i^{th}$ proton, $R_1$ the site-specific local field relaxation rate in the fast motional regime $\gamma_H B \tau_c \ll 1$, and $R_1^{inf}$ the magnetic field-independent relaxation rate. Since the dominant relaxation mechanism of protons is the modulation of dipole-dipole coupling, the $\tau_c^i$ values are expected to be close to each over for a molecule with rigid structure due to overall molecular tumbling; the deviation of $\tau_c^i$ from the average value highlights the molecular sites with increased or decreased mobility with respect to average. Lines in Fig.3B show the best fit of FAD proton NMRD in the region from 0.56 T to 16.44 T (1.77 T – 16.44 T for A8 proton) by expression 1. The extracted correlation times are shown in Fig.3C. The correlation times for protons in the isoalloxazine and adenine rings are alike, especially the ones of protons A8 and F6, meaning that no significant relative motions occur. Similar $\tau_c^i$ values were obtained for the protons of the linker. This observation supports conclusions drawn in recent quantum chemistry calculations of the FAD conformation in water (Kieninger et al., 2020), where a stacked conformation of the adenine and isoalloxazine rings was found in the FAD-water complex.

It is worth noting, that only the "high field" part of the NMRD curve, which corresponds to the transition between the motional regimes, can be used to determine correlation times of individual protons with atomic resolution. In the "low field" part of NMRD where the extreme narrowing condition $\gamma_H B \tau_c \ll 1$ is met, the relaxation time of a particular spin cannot be obtained when spins are strongly coupled. Weak coupling means that the difference in resonance frequencies of a given nucleus $i$ and other nucleus $j$, $|\Omega_i - \Omega_j|$ is larger than the scalar coupling $|J_{ij}|$ between them: $|\Omega_i - \Omega_j| \gg |J_{ij}|$, in the strong coupling of states the inequality is opposite. It was shown previously by measuring proton relaxation dispersion of adenosine monophosphate, AMP, that A8 and A2 protons are strongly coupled (Kiryutin et al., 2016) at magnetic field below 10 mT, although their scalar coupling constant (0.25 Hz) was not observed by line splitting.

In the time-resolved CIDNP spectra (Fig.5) of FAD in aqueous solution obtained without delay after a short laser pulse of 10 ns with detection using an RF-pulse of 1 µs duration signals from adenine and flavin are seen that have remarkably different linewidth. The absorptive lines of the adenine A8 and A2 protons are not as sharp as in the NMR spectrum. For flavin, only a
very broad signal of low intensity is detected in the aliphatic part at the position of the methyl protons that we attributed to formation of the reduced flavin moiety, FADH$^-$. In contrast, in the spectrum obtained with addition of 10 mM of $H_2O_2$ all lines are as sharp as in the ordinary NMR spectrum because addition of 10 mM hydrogen peroxide as a strong oxidizing agent significantly accelerates FADH$^-$ reoxidation to FAD. The absorptive signals of the F7 and F8-methyl protons as well as an emissive CIDNP signal of F9 are seen in the geminate spectrum. The sign of the CIDNP signals are in accordance with
Kaptein's rule for triplet precursor multiplicity and a g-factor of the flavin radical being larger than that of the adenine radical.

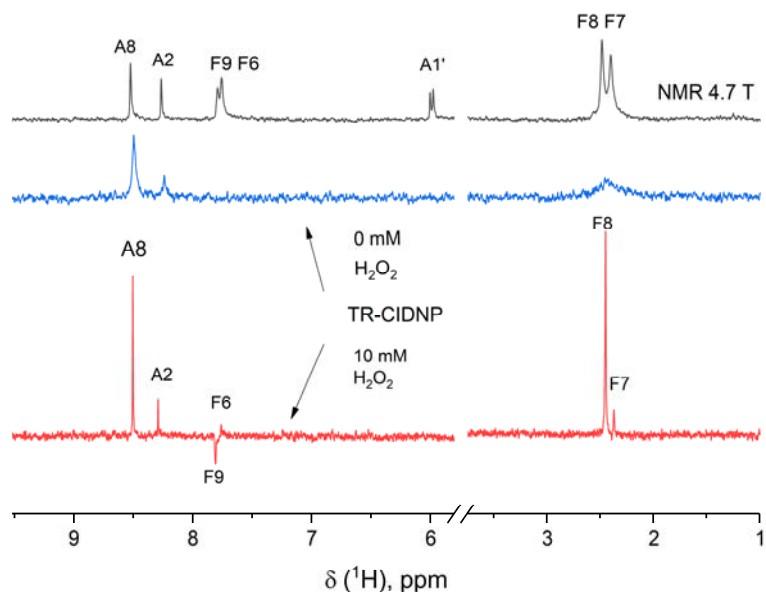

**Fig.5 200-MHz NMR (top) and photo-CIDNP spectra (middle, bottom) of a 0.9 mM solution of FAD in $D_2O$ at pH 3.2 without $H_2O_2$ (middle) and with 10 mM $H_2O_2$ (bottom) taken without delay after a short single laser pulse. Background magnetization is removed by a saturation pulse sequence applied prior to the short single laser pulse. Absorptive CIDNP signals are detected for the H8 and**
**H2 protons of the adenine moiety in both CIDNP spectra, while for the flavin moiety, absorptive signals of the F7 and F8-methyl protons and an emissive CIDNP signal of F9 are seen in the geminate spectrum obtained with addition of $H_2O_2$ (bottom).**

The most intense signal in the geminate $^1H$ CIDNP spectra is the A8 proton signal (Stob et al., 1989), highlighting that the largest spin density in the short-lived charge separated state of FAD is located on this proton. This observation is in agreement with the adenosine cation radical structure and the predicted isotropic hyperfine interaction constants, –0.57 mT for A8 and –
0.32 mT for A2 (Adhikary et al., 2008). For short-lived radical pairs in a non-viscous solvent proportionality between hyperfine coupling constants and geminate CIDNP signal intensities has been established (Morozova et al., 2011). We checked the proportionality of HFC constants and CIDNP for the A8 and A2 adenine protons of FAD (requiring CIDNP being zero for zero HFC) and found full agreement.

In the CIDNP spectra detected under cw-illumination at low magnetic field all signals are emissive (Fig.6B). The position of
the emissive maximum is common for adenine and flavin, the sign of polarization does not depend on the sign of HFC constants. This is in full accordance with the T$_-$-S mechanism of CIDNP.

The main advantage of the fully automated set-up for shuttling the sample is the possibility of fine tuning the experimental conditions. As we noticed, the intensity ratio of the signals from A8 and A2 strongly depends on the irradiation time. This

results from polarization transfer between them and different relaxation times. Since the HFC constant of A8 is larger than that of A2 we opted to measure the CIDNP field dependence of the A8 proton.

However, the relaxation dispersion measurements have shown a very short relaxation time of proton A8 (see Fig.3B) at a magnetic field of 4 mT which is optimal for CIDNP formation. In addition, in this field protons A8 and A2 are strongly coupled in the same way as it was shown for adenosine monophosphate (Kiryutin et al., 2016). Thus, measurement and analysis of field dependent CIDNP of FAD protons should be done with taking into account these circumstances. Although their spin-spin coupling constant is small (below 0.4 Hz), comparable with the linewidths of adenine and is not seen as a splitting, the low B-field gives rise to strong coupling between protons A8 and A2 in FAD and thus leads to coherent transfer of light-induced proton hyperpolarization between them. Since proton A2 has a more than two times longer $T_1$ this proton shows a higher CIDNP effect in comparison to A8 when irradiation time $\tau_{irr}$ is 2-3 seconds. To avoid polarization transfer we used a short irradiation time ($\tau_{irr}$ = 0.5 seconds), which is long enough for A8 to reach its steady-state polarization level, but sufficiently short to keep the share of polarization leaked to A2 relatively small. With such optimized settings, we measured proton CIDNP field dependences of FAD samples at pH 2.7 and pH 3.9. These measured CIDNP field dependences were corrected to the genuine CIDNP field dependence using the relevant nuclear spin relaxation dispersion data and the time profile of the sample transfer. No difference between A8 CIDNP field dependences was found within experimental error except for a four-fold decrease in CIDNP intensity when the pH was changed from 2.7 to 3.9. Further pH increase leads to diminishing CIDNP.

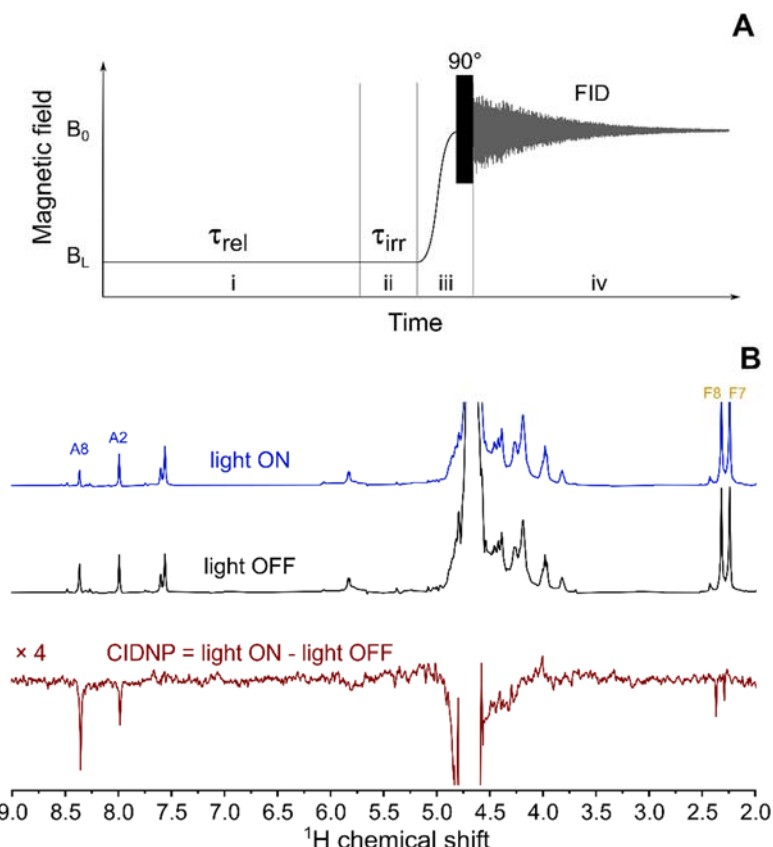

**Fig.6 A** Experimental protocol for measuring the CIDNP field dependence. In stage i the sample relaxes in low field $B_L$ for $\tau_{rel}$ = 5 s. Then, in stage ii the LED is either switched on (in "light ON" experiment) or stays switched off (in "light OFF" experiment) for $\tau_{irr}$ = 0.5 s. In stage iii the sample is transferred to high field $B_0$ and a 90° RF pulse is applied. In stage iv nuclear spin free induction

 **decay is acquired. B Top spectrum: 400 MHz ¹H NMR spectrum obtained using the protocol shown in subplot (A), with 3W 520 nm LED switched on during stage ii of the protocol, $B_L$ = 4 mT; middle spectrum: 400 MHz ¹H NMR spectrum obtained using the protocol shown in subplot (A), LED is switched off during stage ii of the protocol, $B_L$ = 4 mT; bottom spectrum: $B_L$ = 4 mT CIDNP spectrum which is the difference between "light ON" and "light OFF" spectra taken in this field.**

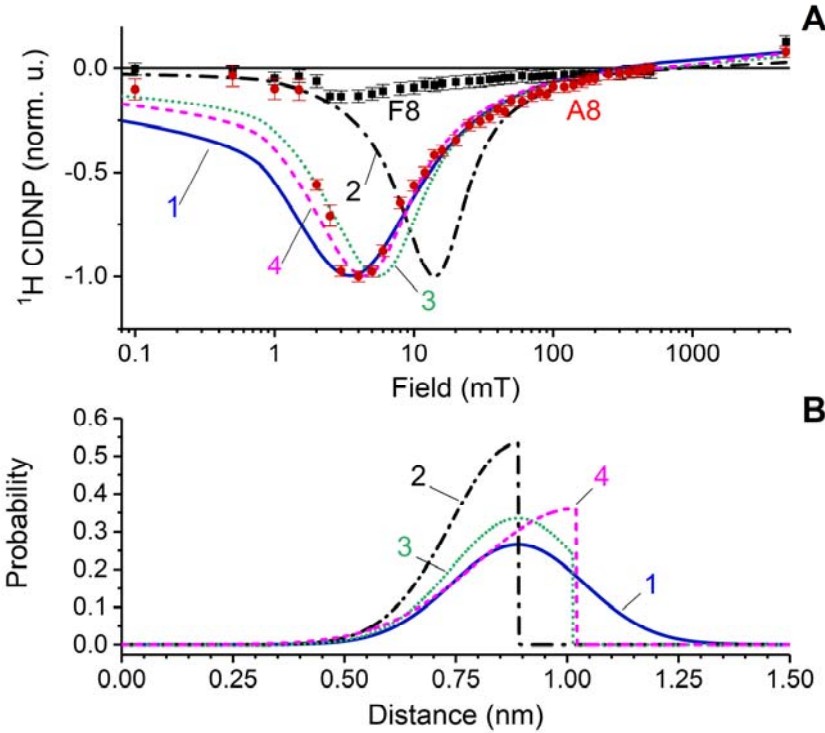

**Fig.7 A. CIDNP dependence on the magnetic field. Red circles and black squares – experimental data for adenine A8 and flavin F8 protons; CIDNP data corrected by taking into account nuclear relaxation occurring during sample transfer to the detection position at high field. Lines show the numerical simulation of CIDNP field dependences with the parameters listed in Table 1, using four different distance distribution functions which are depicted in the subplot B. The fit quality is characterized by the q-value that is the sum of squared deviations from the experimentally observed data. The q values are $1.8·10^{-3}$, $9·10^{-3}$, $5.2·10^{-3}$ and $3.9·10^{-3}$ for**
**simulations 1-4, respectively. B Model biradical end-to-end distribution functions used for simulations of the CIDNP field dependence: Simulation 1 (blue solid line) – normal distribution centered at 0.89 nm with standard deviation 0.15 nm; Simulation 2 (black dash-dot line) – left half of normal distribution centered at 0.89 nm with standard deviation 0.15 nm; Simulation 3 (green dot line) – normal distribution centered at 0.89 nm with standard deviation 0.15 nm, with distances above 1.03 nm cut; Simulation 4 (magenta dash line) – left half of normal distribution centered at 10.2 nm with standard deviation 0.22 nm.**

The CIDNP data for proton A8 are shown in figure Fig.7A by red circles. A well pronounced single maximum is detected in the wide range of magnetic fields between 0.1mT and 9.4 T. The high quality of the data left no doubts that only one maximum of CIDNP is detected in the field dependence excluding that two types of biradical with different exchange interaction are formed from FAD. The maximum is located at 4 mT, the full width of the maximum is about 10 mT.

To get more detailed information about the structure and exchange interaction in the FAD biradical, we simulated the CIDNP
field dependence using the approach originally proposed in the work of Kaptein and co-authors (de Kanter et al., 1977) for calculating CIDNP in flexible biradicals. This model was widely used in our studies of CIDNP in cyclic aliphatic ketones (Tsentalovich et al., 2002; Yurkovskaya et al., 1995; Morozova et al., 1997a; Morozova et al., 1997b; Tsentalovich et al., 1997) and the model compound containing flavin and tryptophan connected by a polymethylene chain (Paul et al., 2017).

The position of the CIDNP maximum and the shape of the simulated CIDNP field dependence strongly depend on the radial distribution function chosen. Based on both theoretical and experimental evidence of closed and rigid structures of FAD in aqueous solution we assume that the light-induced biradical state of FAD conserves these properties to a large extent. We tested several model functions of radial distribution, and found the best correspondence between simulation and experiment for a normal distribution, centered at $r_0 = 0.89$ nm, with standard deviation $\sigma = 0.15$ nm. We also tried other end-to-end distribution functions for simulation of CIDNP field dependences. The results are shown in Fig.7. Simulation 1 (blue solid line) – normal distribution centered at 0.89 nm with a standard deviation of 0.15 nm; Simulation 2 (black dash-dot line) – left half of a normal distribution centered at 0.89 nm with standard deviation of 0.15 nm; Simulation 3 (green dot line) – normal distribution centered at 0.89 nm with standard deviation of 0.15 nm, but distances above 1.03 nm are cut; Simulation 4 (magenta dash line) – left half of a normal distribution centered at 10.2 nm with a standard deviation of 0.22 nm. The fit quality is characterized by the q-value that is the sum of squared deviations from the experimentally observed data which are $1.8 \cdot 10^{-3}$, $9 \cdot 10^{-3}$, $5.2 \cdot 10^{-3}$ and $3.9 \cdot 10^{-3}$ for simulations 1-4, respectively. Thus we conclude that the best function is a normal distribution centered at 0.89 nm with a standard deviation of 0.15 nm.

We also preliminarily examined samples containing FAD with addition of tryptophan at variable concentration. For low tryptophan concentration, the emissive maximum at low field stayed at its position around 4 mT for flavin and adenine protons, while at the high field polarization of tryptophan was observed. It is a clear indication that the spin evolution of the biradical has an influence on the overall magnetic field dependence of the flavin radical in the primary biradical and in the secondary radical pair where spins are not correlated. We detected a remarkably different dependence of CIDNP for flavin and tryptophan under variation of the tryptophan concentration, but a detailed discussion is beyond the scope of the present paper. This work is underway in our laboratories and the results will be published elsewhere.

**Table 1.** Parameters used to model the CIDNP field dependence:

| Symbol | Description | Value |
|---|---|---|
| $g_a$ | g-factor of the first radical (adenine) | 2.0034 |
| $A$ | HFI constant on spin-1/2 nuclei to be observed | –0.7 mT |
| $g_b$ | g-factor of the second radical (flavin) | 2.0035 |
| $J_0$ | amplitude parameter of exchange interaction, $J_{ex}(r)=J_0 \cdot e^{-\alpha r}$ | $-2.3 \cdot 10^8$ mT |
| $\alpha$ | exchange interaction distance decay parameter | 0.214 nm |
| $D$ | effective radial diffusion coefficient in biradical state | $2 \cdot 10^{-7}$ cm$^2$/s |
| $G$ | mean-square fluctuating local field | $6.1 \cdot 10^{17}$ s$^{-2}$ |
| $\tau_u$ | local field correlation time | 1 ps |
| $\tau_{rot}$ | rotational diffusion correlation time | 800 ps |
| $k_p$ | recombination rate constant from singlet state | $2 \cdot 10^{10}$ s$^{-1}$ |
| $k_s$ | scavenging rate to minor reaction products | $10^5$ s$^{-1}$ |
| $A_{add}$ | HFI constant with additional spin-1/2 nuclei | 1.67 mT |
| $n$ | number of additional nuclei | 4 |

**Conclusions**

In this study we measured and analyzed the magnetic field dependence of $^1$H-CIDNP to confirm the involvement of the adenine radical in the primary photochemical reaction of intramolecular electron transfer in FAD resulting in formation of the flavin-adenine biradical. By reducing the light irradiation time to 0.5 s for CIDNP formation at low magnetic field we avoided coherent polarization transfer among the protons of adenine and obtained for the A8 proton a magnetic field dependence with a single emissive maximum located at 4 mT. A dependence of the same shape was detected for the methyl protons of flavin. The dependence of relaxation times $T_1$ on the magnetic field between 1 and 16 T allowed us to determine the correlation times $\tau_c$ of intramolecular mobility with atomic resolution. From the coincidence of $\tau_c$ for the protons of flavin and adenine and the absence of any short correlation times we conclude that the structure of the FAD molecule is rigid.

Modeling of the CIDNP field dependence in frame of the model proposed by Kaptein provides good agreement with the experimental data for a normal distance distribution between the two radical centers of 0.89 nm with a standard deviation of 0.15 nm. Time-resolved CIDNP spectra recorded without delay between the short laser excitation pulse and detection confirmed that back electron transfer leads to formation of a diamagnetic adenine and reduced flavin of $FADH^-$, whereas with addition of oxidizing agent $H_2O_2$ the diamagnetic FAD is restored on the geminate stage.

**Author contribution**

AVY and HMV designed the project. IVZ and NNF conducted the CIDNP experiments, ASK measured the proton nuclear relaxation dispersion. MSP performed the quantum chemistry calculations. NNL and KLI wrote the CIDNP simulation code. IVZ conducted the numerical simulations of the CIDNP field dependence. IVZ, NNF, OBM, HMV, RZS and AVY analysed the results. All authors contributed to manuscript preparation.

**Competing interests**

The authors declare that they have no conflict of interest.

**Acknowledgements**
The experimental part of this work has been supported by the Russian Science Foundation (project 20-63-46034) while the theoretical part got support from a joint RFBR-DFG grant (project #21-53-12023).

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
