# Peer review of "Exchange interaction in short lived flavine adenine dinucleotide biradical in aqueous solution revisited by CIDNP and nuclear magnetic relaxation dispersion"

_Magnetic Resonance, 2021_

## Referee Comment (RC1)

**Exchange interaction in short lived flavin adenine dinucleotide biradical in aqueous solution revisited by CIDNP and nuclear magnetic relaxation dispersion**

*Zhukov, Kiryutin, Panov, Fishman, Morozov, Lukzen, Ivanov, Vieth, Sagdeev, Yurkovskaya*

A nice manuscript. Publishable after minor corrections.

Scheme 1 appears on page 2 but is not referred to until page 7. Perhaps it should be moved nearer to the place where it's actually needed.

Lines 35-36: it is an exaggeration to say that the radical-pair mechanism is "generally accepted" as the "explanation of bird navigation". Although it is clear that migratory birds have a magnetic compass sense, they also use the sun, stars, olfaction, landmarks etc. Additionally, the RPM hypothesis of magnetoreception is not yet "generally accepted".

Lines 41-42: "In a recently discovered DNA based magnetic sensor FAD was used to repair a DNA lesion by splitting a thymine dimer (Zwang et al., 2018)". This is a curious way to refer to DNA photolyase. I think the authors should at least say that $FADH^-$, the fully reduced form of FAD, acts as the chromophore and electron donor in this light-dependent DNA repair enzyme. It is also far from clear that there should be any magnetic field effect on this reaction (*ACS Cent. Sci.*, **4** (2018) 318-320).

I think the following two articles (at least) should be cited to avoid giving the impression that no-one other than Stob *et al*. (1989) has studied magnetic field effects on FAD photochemistry.

Murakami, M., K. Maeda, and T. Arai. 2005. Dynamics of intramolecular electron transfer reaction of FAD studied by magnetic field effects on transient absorption spectra. *J. Phys. Chem. A* **109**:5793-5800.

Antill, L. M., and J. R. Woodward. 2018. Flavin adenine dinucleotide photochemistry is magnetic field sensitive at physiological pH. *J. Phys. Chem. Lett.* **9**:2691-2696.

Lines 63-64: I was puzzled by the mention of tryptophanyl radicals here. This only became clear in line 98. A few more words of explanation, or references to studies of FAD-Trp photo-reactions, would be helpful.

Line 190: it would be useful to have either a literature reference for this equation or some description of how it was derived.

Lines 211-213: why does hydrogen peroxide cause the spectra to become sharper? Is this because it suppresses the comproportionation reaction, $FAD + FADH^- + H^+ \rightarrow 2FADH^\bullet$ ?

Lines 249-251: is the pH dependence of the CIDNP intensities evidence for a change in the stacking of the F and A groups?

Lines 272-274: "The high quality of the data left no doubts that only one maximum of CIDNP is detected in the field dependence excluding that two types of biradical with different exchange interaction are formed from FAD." What evidence do the authors have that there are not, in fact, two unresolved peaks? If there were, their results would not disagree quite so much with Stob, Kemminck and Kaptein.

Table 1: should the units of $G$ be $s^{-2}$ ?

---

## Author Comment (AC1)

**Response to Prof. Peter Hore Comments**

We thank Prof. Peter Hore for the careful reading of the manuscript and the high estimation of our results. Our response to the individual points of criticism is found below: while the reviewer's statements are printed in italics, our comments and changes are printed in standard font.

*A nice manuscript. Publishable after minor corrections.*
*Scheme 1 appears on page 2 but is not referred to until page 7. Perhaps it should be moved nearer to the place where it's actually needed.*

We have corrected this inaccuracy and referred Scheme 1 on page 2, when a study of Robert Kaptein and co-workers (Stob et al., 1989) is described.

*Lines 35-36: it is an exaggeration to say that the radical-pair mechanism is "generally accepted" as the "explanation of bird navigation". Although it is clear that migratory birds have a magnetic compass sense, they also use the sun, stars, olfaction, landmarks etc. Additionally, the RPM hypothesis of magnetoreception is not yet "generally accepted".*

This sentence has been rephrased as "A review on the radical-pair mechanism (RPM) of magnetoreception as a leading hypothesis to explain the bird navigation can be found in the literature (Hore and Mouritsen, 2016)".

*Lines 41-42: "In a recently discovered DNA based magnetic sensor FAD was used to repair a DNA lesion by splitting a thymine dimer (Zwang et al., 2018)". This is a curious way to refer to DNA photolyase. I think the authors should at least say that $FADH^-$, the fully reduced form of FAD, acts as the chromophore and electron donor in this light-dependent DNA repair enzyme. It is also far from clear that there should be any magnetic field effect on this reaction (ACS Cent. Sci., 4 (2018) 318-320).*

We have deleted this sentence for clarity.

*I think the following two articles (at least) should be cited to avoid giving the impression that no-one other than Stob et al. (1989) has studied magnetic field effects on FAD photochemistry.*

*Murakami, M., K. Maeda, and T. Arai. 2005. Dynamics of intramolecular electron transfer reaction of FAD studied by magnetic field effects on transient absorption spectra. J. Phys. Chem. A 109:5793-5800.*

*Antill, L. M., and J. R. Woodward. 2018. Flavin adenine dinucleotide photochemistry is magnetic field sensitive at physiological pH. J. Phys. Chem. Lett. 9:2691-2696.*

The articles mentioned above have been cited (Line 49).

*Lines 63-64: I was puzzled by the mention of tryptophanyl radicals here. This only became clear in line 98. A few more words of explanation, or references to studies of FAD-Trp photo-reactions, would be helpful.*

We extended the paragraph and wrote: Often FAD is discussed as a candidate molecule responsible for the formation of such spin-correlated radical pairs in living organisms that contain particular proteins - blue light photoreceptors, cryptochromes, which contain a non-covalently bound FAD photoreceptor molecule. The radical pair usually considered is a pair of radicals [FAD$^{\bullet-}$ Trp$^{\bullet+}$], which is formed by sequential electron transfer along the chain of tryptophan residues to the cofactor FAD in cryptochrome (Dodson et al., 2015). However, the appearance of the magnetic field effect in this secondary radical pair, [FAD$^{\bullet-}$ Trp$^{\bullet+}$], formed in parallel or subsequently from the FAD biradical might be significantly affected by the spin dynamics in the primary FAD biradical.

*Line 190: it would be useful to have either a literature reference for this equation or some description of how it was derived.*

We have added the following sentences describing derivation of the equation 1. "assuming a simple empirical model with two contributions to relaxation, one resulting from site-specific local field correlation time $\tau_c^i$ and another one being a field-independent constant. Accordingly, the total relaxation rate is given by the sum $R_1^{tot}(B) = \frac{R_1}{1+(\gamma_H B \tau_c)^2} + R_1^{inf}$". " Since the dominant relaxation mechanism of protons is the modulation of dipole-dipole coupling, the $\tau_c^i$ values are expected to be close to each over for a molecule with rigid structure due to overall molecular tumbling; the deviation of $\tau_c^i$ from average value highlights the molecular sites with increased or decreased mobility with respect to average. "

*Lines 211-213: why does hydrogen peroxide cause the spectra to become sharper? Is this because it suppresses the comproportionation reaction, FAD+FADH$^-$+H$^+$→2FADH$^\bullet$?*

Yes, we think so.
We checked that the addition of H$_2$O$_2$ (with ~10 mM concentration) has prevented the photo-bleaching of the FAD sample like it was published in the work of Maeda et al. for FMN as a photosensitizer [Maeda, K., Lyon, C., Lopez, J., Cemazar, M., Dobson, C. & Hore, P.J. Improved photo-CIDNP methods for studying protein structure and folding. J. Biomol. NMR 16, 235–244 (2000)].

*Lines 249-251: is the pH dependence of the CIDNP intensities evidence for a change in the stacking of the F and A groups?*

R. Kaptein studied the dependence of photo-CIDNP and fluorescence of FAD on the pH in his work (Stob et al., 1989). These two dependences are astonishingly similar. A maximum of both CIDNP and fluorescence was observed at pH=2.4 with a sharp decrease at both low and high pH.

In our turn we have also checked the dependence of geminate CIDNP (CIDNP spectra were taken without delay after a short single laser pulse) of FAD on the pH of aqueous solution (see Fig. 1 below), and obtained curves that are very close to Kaptein's dependences.

[Figure]

**Fig.1.** The pH-dependence of the geminate CIDNP intensity of adenine protons A2 (red) and A8 (black) of a 0.9 mM solution of FAD in $D_2O$; CIDNP spectra were taken without delay after a short single laser pulse by 64 scans; a 6 µs RF pulse was used for detection.

In Kaptein's work (Stob et al., 1989) there is the following reasoning why the CIDNP was observed only at narrow pH range: "the decrease of CIDNP in the high pH region is not due to a low yield of triplet flavins but is an inherent property of the biradical nature of the effect... In the low pH region the reduction of CIDNP intensity follows that of the fluorescence quenching...it is likely to be due to radiationless decay of the flavin excited singlets with concomitant reduction of the flavin triplet yield." "the fluorescence quenching at neutral pH in fact reflects the conformational equilibrium of ground-state FAD, since the lifetime of the singlet-state flavin is too short to allow for an equilibrium to be attained. We explain the pH-dependent CIDNP behavior also on the basis of a ground-state property of FAD, in this case the pKa of 3.6 of the adenine moiety. The reason is that now the protonation-deprotonation equilibria are slow compared to the CIDNP time scale of $10^{-7}-10^{-9}$ s (except for intramolecular proton transfer) so that the pH dependence is governed by the protonation state of the precursor molecule (FAD). The complete suppression of CIDNP at neutral pH indicates that the stacking equilibrium in the FH$^\bullet$-A$^\bullet$ biradical is shifted completely to the stacked form before T-S mixing can occur, in contrast to that of FAD itself where open conformations are present for about 20%."

*Lines 272-274: "The high quality of the data left no doubts that only one maximum of CIDNP is detected in the field dependence excluding that two types of biradical with different exchange interaction are formed from FAD." What evidence do the authors have that there are not, in fact, two unresolved peaks? If there were, their results would not disagree quite so much with Stob, Kemminck and Kaptein.*

We have made the following simulations suggested by P.J. Hore and referee #2, see Fig.4. We think it is clear from this figure that a second "open" conformation should give CIDNP at fields lower than 1 mT (at least in simulations), but we did not observe it in our experiments.

Here, numbers correspond to the following simulations:

1: normal distribution with its center at $r_0 = 0.89$ nm and a standard deviation $\sigma = 0.15$ nm - same as simulation 1 in main text;

2: normal distribution with $r_0 = 1.48$ nm and $\sigma = 0.15$ nm;

3: composite distribution made of equally weighted distributions used in sim.1 and sim.2;

4: composite distribution made of equally weighted distributions: a) $r_0 = 0.75$ nm $\sigma = 0.05$ nm, b) $r_0 = 1$ nm $\sigma = 0.05$ nm;

Apparently, the single maximum distribution used in sim.1 gives the best fit to the data.

[Figure]

Fig. 4. Top – CIDNP field dependences: black squares – A8 experimental data, lines – simulation results for the model distributions 1-4 normalized to the maximum absolute experimental CIDNP intensity. Middle – distribution functions used in simulations 1-4. Bottom – calculated CIDNP field dependences for simulations 1-4 without normalization.

The simulations also presented in the answer to the Referee #2 comment (iv) show that models with two distinct biradical conformations lead to a pronounced deviation of the simulation results from the observed CIDNP field dependence. Although this does not mean that deviation of the biradical end-to-end distribution from a normal distribution is excluded, the merit of such deviation is anticipated to be relatively small. We believe that this allows us to talk about one type of the biradical conformation.

*Table 1: should the units of G be $s^{-2}$?*

Yes, it was a typo, there should be $s^{-2}$.

We hope that our manuscript in its present form is suitable for publication in the special issue of Magnetic Resonance dedicated to 80$^{th}$ anniversary of Professor Kaptein.

---

## Author Comment (AC2)

**Response to Comments of Anonymous Referee #2**

We thank the reviewer for her/his thorough evaluation of our manuscript and the valuable suggestions for improvement. Our response to the individual points of criticism is found below: while the reviewer's statements are printed in italics, our comments and changes are printed in standard font.

*The very well written and detailed manuscript by Zhukov et al. by the Yurkovskaya group studied the field-dependent photo-CIDNP activity and 1H relaxation by a field cycling device of Flavin adenine dinucleotide (FAD) which is an important cofactor in many light-sensitive enzymes. The very high resolution field-dependent data collected indicates the presence of only a single FAD biradical attributed to the closed conformation. The following suggestions are indicated*

*(i) Line 205 remove the word "of"*

We removed the word "of" at line 205.

*(ii) The measurements were done at very low pHs 2.7 and 3.9, which is far away from physiological pH. In addition, the strong CIDNP activity vanishes above the pH range measured. Possible reasons for the loss of CIDNP activity should be discussed. Furthermore, can the authors exclude that their findings on a single biradical hold at neutral pH? Possibly measurements on the 1H relaxation dispersion at physiological pH and comparison to the acidic pH data may give insights.*

R. Kaptein studied the dependence of photo-CIDNP and fluorescence of FAD on the pH in his work (Stob et al., 1989). These two dependences are astonishingly similar. A maximum of both CIDNP and fluorescence was observed at pH=2.4 with a sharp decrease at both low and high pH.

In our turn we have also checked the dependence of geminate CIDNP (CIDNP spectra were taken without delay after a short single laser pulse) of FAD on the pH of aqueous solution (see Fig. 1 below), and obtained curves that are very close to  Kaptein's dependences.

In Kaptein's work (Stob et al., 1989) there is the following reasoning why the CIDNP was observed only at narrow pH range: "the decrease of CIDNP in the high pH region is not due to a low yield of triplet flavins but is an inherent property of the biradical nature of the effect… In the low pH region the reduction of CIDNP intensity follows that of the fluorescence quenching…it is likely to be due to radiationless decay of the flavin excited singlets with concomitant reduction of the flavin triplet yield." "the fluorescence quenching at neutral pH in fact reflects the conformational equilibrium of ground-state FAD, since the lifetime of the singlet-state flavin is too short to allow for an equilibrium to be attained. We explain the pH-dependent CIDNP behavior also on the basis of a ground-state property of FAD, in this case the pKa of 3.6 of the adenine moiety. The reason is that now the protonation-deprotonation equilibria are slow compared to the CIDNP time scale of $10^{-7}$-$10^{-9}$ s (except for intramolecular proton transfer) so that the pH dependence is governed by the protonation state of the precursor molecule (FAD). The complete suppression of CIDNP at neutral pH indicates that the stacking equilibrium in the FH$^\bullet$-A$^\bullet$ biradical is shifted completely to the stacked form before T-S mixing can occur, in contrast to that of FAD itself where open conformations are present for about 20%."

We also have measured $^1$H nuclear spin relaxation dispersion by the inversion-recovery method of a 4.8 mM solution of FAD in $D_2O$ pH 6.6 (see Fig. 2 below) as was suggested by **Referee #2**. The

data show that the local field correlation times of adenine and isoalloxazine rings do no longer coincide; thus we conclude that at neutral pH FAD adopts a conformation with higher relative mobility of these aromatic rings, which is drastically different from the stacked one in the acidic pH range.

[Figure]

**Fig.1.** The pH-dependence of the geminate CIDNP intensity of adenine protons A2 (red) and A8 (black) of a 0.9 mM solution of FAD in $D_2O$; CIDNP spectra were taken without delay after a short single laser pulse by 64 scans; a 6 µs RF pulse was used for detection.

[Figure]

**Fig.2.** Nuclear spin relaxation dispersion taken by the inversion–recovery method for six selected protons of FAD: A8 – black squares, A2 – red circles, F6 – blue up triangles, F5',5''– magenta diamonds, F9 – green down triangles. of a 4.8 mM solution of FAD in D$_2$O pH 6.6, 27°C.

*(iii) The data points in Figure 7A at low field drop much faster than theoretically predicted. Reasons for this apparent discrepancy should be discussed. In this context, it would be interesting for the reader to see the importance/extend of the correction of the CIDNP data by taking into account nuclear relaxation occurring during sample transfer by showing in a Suppl. Figure non corrected data. Furthermore, it is suggested to show on the y-axis not normalized units, but signal enhancement in respect to "no light".*

The extent of the CIDNP field dependence correction is shown in Fig.3a. Black squares – measured CIDNP intensity dependence on the sample irradiation field $B_L$, red circles – CIDNP values which are calculated from measured one taking into account A8 proton spin relaxation during sample transfer to the observation position at high field $B_0$. The sample spends most of the transfer time at field region with low A8 T1 relaxation (Fig.3b). To measure CIDNP at $B_L < 2$ mT we have to transfer sample to the external electromagnet outside the fringe field of the NMR spectrometer superconducting magnet. This leads to ca. 30% increase of transfer time and additional polarization losses during sample transfer. Nevertheless, the CIDNP intensity correction on relaxation during sample transfer works well, as is marked by the asterisk for the data taken at $B_L = 2.5$ mT when the experiment has been done in the external electromagnet. Here, the raw CIDNP intensity deviates from the smooth curve, whereas after correction the deviation extent is negligible.

In our opinion, the discrepancy between simulated and observed CIDNP at low field results from the fact that the biradical end-to-end distribution function is not fully optimal. Due to decaying exponential dependence of the exchange interaction, the biradical conformations with larger distances between radical centers contribute to CIDNP at lower fields. While giving good approximation at fields $B_L > 3$ mT, the normal distribution used in sim.1 fails to fit the low field CIDNP data. The correspondence between simulation and experiment at low field improves then using distribution from sim.4, but the overall correspondence decreases. We think that using automatic algorithms of optimal distribution function search it is possible to get a nice fit of all

data, but the development of such an algorithm is beyond of the scope of this work.

[Figure]

**Fig. 3.** (a) Black squares – measured CIDNP intensities dependence on sample irradiation field $B_L$, red circles – CIDNP data after correction on polarization losses during the sample transfer. (b) The time profile of field variation during the sample transfer from $B_L = 4$ mT to the observation field $B_0 = 9.4$ T.

*(iv) The authors state that the measured data indicate the presence of only a single biradical. How sensitive is the method presented? It is suggested to make calculations with two states (for example state A and B from Figure 4) at different ratios.*

We have made the following simulations suggested by P.J. Hore and referee #2, see Fig.4. We think it is clear from this figure that a second "open" conformation should give CIDNP at fields lower than 1 mT (at least in simulations), but we did not observe it in our experiments.

Here, numbers correspond to the following simulations:

1: normal distribution with its center at $r_0 = 0.89$ nm and a standard deviation $\sigma = 0.15$ nm - same as simulation 1 in main text;

2: normal distribution with $r_0 = 1.48$ nm and $\sigma = 0.15$ nm;

3: composite distribution made of equally weighted distributions used in sim.1 and sim.2;

4: composite distribution made of equally weighted distributions: a) $r_0 = 0.75$ nm $\sigma = 0.05$ nm, b) $r_0 = 1$ nm $\sigma = 0.05$ nm;

Apparently, the single maximum distribution used in sim.1 gives the best fit to the data.

[Figure]

Fig. 4. Top – CIDNP field dependences: black squares – A8 experimental data, lines – simulation results for the model distributions 1-4 normalized to the maximum absolute experimental CIDNP intensity. Middle – distribution functions used in simulations 1-4. Bottom – calculated CIDNP field dependences for simulations 1-4 without normalization.

We hope that our manuscript in its present form taking into account the additional data shown in the rebuttal letter is suitable for publication in the special issue of Magnetic Resonance dedicated to 80[th] anniversary of Professor Kaptein.